# Phenome-Wide Analysis of Coffee Intake on Health over 20 Years of Follow-Up Among Adults in Hong Kong Osteoporosis Study

**DOI:** 10.3390/nu16203536

**Published:** 2024-10-18

**Authors:** Jonathan K. L. Mak, Yin-Pan Chau, Kathryn Choon-Beng Tan, Annie Wai-Chee Kung, Ching-Lung Cheung

**Affiliations:** 1Department of Pharmacology and Pharmacy, The University of Hong Kong, Hong Kong, China; jklmak@hku.hk (J.K.L.M.); ypchau@hku.hk (Y.-P.C.); 2Department of Medical Epidemiology and Biostatistics, Karolinska Institutet, 17177 Stockholm, Sweden; 3Department of Medicine, School of Clinical Medicine, The University of Hong Kong, Hong Kong, China; kcbtan@hku.hk (K.C.-B.T.); awckung@hku.hk (A.W.-C.K.); 4Laboratory of Data Discovery for Health (D^2^4H), Hong Kong Science Park, Pak Shek Kok, Hong Kong, China

**Keywords:** coffee, PheWAS, health outcomes, mortality, prospective cohort study

## Abstract

Background/Objectives: There has been limited evidence on the long-term impacts of coffee intake on health. We aimed to investigate the association between coffee intake and the incidence of diseases and mortality risk over 20 years among community-dwelling Chinese adults. Methods: Participants were from the Hong Kong Osteoporosis Study who attended baseline assessments during 1995–2010. Coffee intake was self-reported through a food frequency questionnaire and was previously validated. Disease diagnoses, which were mapped into 1795 distinct phecodes, and mortality data were obtained from linkage with territory-wide electronic health records. Cox models were used to estimate the association between coffee intake and the incidence of each disease outcome and mortality among individuals without a history of the respective medical condition at baseline. All models were adjusted for age, sex, body mass index, smoking, alcohol drinking, and education. Results: Among the 7420 included participants (mean age 53.2 years, 72.2% women), 54.0% were non-coffee drinkers, and only 2.7% consumed more than one cup of coffee per day. Over a median follow-up of 20.0 years, any coffee intake was associated with a reduced risk of dementia, atrial fibrillation, painful respirations, infections, atopic dermatitis, and dizziness at a false discovery rate (FDR) of <0.05. Furthermore, any coffee intake was associated with an 18% reduced risk of all-cause mortality (95% confidence interval = 0.73–0.93). Conclusion: In a population with relatively low coffee consumption, any coffee intake is linked to a lower risk of several neurological, circulatory, and respiratory diseases and symptoms, as well as mortality.

## 1. Introduction

Coffee is one of the most widely consumed beverages worldwide, and its effects on health remain a topic of ongoing debate [1,2]. Historically, concerns have been raised about the potential risks associated with coffee and caffeine, including conditions like hypertension [3], osteoporotic fractures [4], and anxiety [5]. However, recent research indicated that coffee may also provide various health benefits due to the presence of a range of bioactive compounds in coffee, such as chlorogenic acids, the alkaloid trigonelline, diterpenes (cafestol and kahweol), and melanoidins [6]. For instance, chlorogenic acids in coffee have been linked to anti-oxidant, anti-cancer, and anti-inflammatory effects [7]. Similarly, diterpenes have demonstrated potential pharmacological actions, including hepatoprotective, anti-cancer, anti-diabetic, and anti-osteoclastogenesis properties [8].

Recent epidemiological studies have predominantly suggested a protective effect of coffee consumption against several diseases, including cardiovascular diseases [9,10], type 2 diabetes [11], stroke [12,13], liver diseases [14], depression [15], Parkinson’s disease [16], and cognitive decline and dementia [17,18]. Additionally, coffee has been associated with a reduced risk of specific cancers, such as liver and endometrial cancer [19], as well as all-cause and cardiovascular mortality [20,21,22]. However, recent meta-analyses have also linked excessive coffee intake to an increased risk of hip fracture [23], pregnancy loss [24], and adverse birth outcomes [25]. Notably, previous observational studies have mostly focused on shorter follow-up periods of less than 20 years [2], and there is a scarcity of research comprehensively examining the effects of coffee intake on multiple health outcomes. Moreover, these studies were predominantly conducted in Caucasian populations, where the results could potentially be confounded by factors related to the Western lifestyle, such as smoking [26,27].

Therefore, in the present study, we aimed to investigate the long-term effects of coffee intake on health, over a median follow-up of 20 years, in a cohort of community-dwelling Chinese adults. We first performed a phenome-wide association analysis to identify disease outcomes associated with coffee intake. To validate previous findings, we further performed a targeted assessment of specific disease and mortality outcomes previously linked to coffee intake [2].

## 2. Materials and Methods

### 2.1. Participants

We used data from the Hong Kong Osteoporosis Study (HKOS), which was the first prospective cohort study of osteoporosis and fracture in Asia [28]. Between 1995 and 2010, a total of 9449 southern Chinese women and men were recruited from the community. During baseline assessments, participants completed a questionnaire on lifestyle behaviors, dietary habits, and medical conditions, alongside anthropometric and clinical measurements [28]. Participants were also linked to the Clinical Data Analysis and Reporting System (CDARS), a territory-wide electronic health record database managed by the Hong Kong Hospital Authority, via anonymized unique reference keys. The CDARS captures both inpatient and outpatient records, covering over 80% of hospital admissions in Hong Kong [29]. Electronic health records of the HKOS participants were retrieved from the CDARS until 15 January 2024, allowing us to follow the participants longitudinally with a median of 20 years. Ethics approval of this study was obtained from the Institutional Review Board of The University of Hong Kong/HA HKW, HKSAR, China (reference number UW 23-396). All participants provided informed consent prior to data collection.

For the present analysis, we excluded individuals who were not linked to the CDARS (*n* = 188), with missing demographic data or aged <18 years (*n* = 41), and with missing information on coffee consumption (*n* = 1306) or covariates (*n* = 494), resulting in a sample size of *n* = 7420. Participants with a history of the respective medical condition at baseline were excluded from the analysis of each disease outcome (Appendix A).

### 2.2. Habitual Coffee Intake

Coffee intake was assessed by a self-reported food frequency questionnaire administered during the baseline assessments. Participants were asked about the frequency of their coffee consumption over the past year, as well as the quantity consumed per occasion, measured in cups of 250 mL each. The number of cups consumed per day was then calculated for each individual. Coffee consumption in the HKOS was previously validated through metabolomics analysis [30]. Due to the overall low coffee consumption in our sample (Appendix A), we considered coffee intake as a binary variable (any vs. no coffee intake) and examined the association between any coffee intake and the health outcomes in the analysis.

### 2.3. Health Outcomes

Information on disease diagnoses, coded according to the *International Classification of Diseases*, *Ninth Revision* (ICD-9), as well as mortality was extracted from electronic health records within the CDARS. For the phenome-wide analysis, we defined phenotypes using the “phecode” classification system, which groups related ICD codes into distinct medical conditions and is widely used in phenome-wide association studies [31]. Compared to ICD codes, phecodes are better aligned with the diseases mentioned in clinical practice [32]. Using the R package *PheWAS* [33], we mapped 3347 ICD-9 codes into 1795 unique phecodes and included outcomes with at least 50 cases in the analysis.

We further analyzed 19 specific diseases and two mortality outcomes (all-cause mortality and cardiovascular mortality), selected a priori from the largest umbrella review of coffee consumption and health to date [2]. The list of the selected diseases and the coding method is shown in Appendix A. Cardiovascular mortality was coded based on ICD-10 codes [34], including heart diseases (I00–I09, I11, I13, I20–I51), essential hypertension and hypertensive renal disease (I10, I12, I15), and cerebrovascular diseases (I60–169).

For each health outcome, participants without a history of the respective medical condition were followed from baseline until the first occurrence of the outcome, death, or the end of the study period (15 January 2024), whichever came first.

### 2.4. Statistical Analysis

Participants’ characteristics were summarized as means and standard deviations (SD) for continuous variables and as frequencies and percentages for categorical variables.

A phenome-wide association analysis was first performed using Cox models to identify disease outcomes (phecodes) associated with any coffee intake. We included age, sex, body mass index (BMI), smoking, alcohol drinking, and education as covariates in the models. The Benjamini–Hochberg false discovery rate (FDR) method was applied to account for multiple testing [35], and an FDR < 0.05 was considered as statistically significant.

To validate previous findings, we further assessed the association between any coffee intake and the incidence of the 19 specific diseases and two mortality outcomes selected from the literature [2]. For analysis of all-cause mortality, we plotted Kaplan–Meier curves to compare survival probabilities between non-coffee drinkers and individuals with any coffee intake. Cox models were used to estimate the association between coffee intake and the specific disease and mortality outcomes, adjusted for age, sex, BMI, smoking, alcohol drinking, and education.

For the outcomes associated with coffee intake at FDR < 0.05, we performed subgroup analyses by sex (women vs. men) and age at baseline (<60 vs. ≥60 years) to examine potential effect modification. Two sensitivity analyses were performed. First, to address potential reverse causation, we assessed the association between coffee intake and the identified outcomes by excluding the first two years of follow-up. Second, to examine potential dose–response relationships, we categorized coffee intake into three groups and performed the phenome-wide analysis comparing individuals who consumed ≤1 cup and >1 cup of coffee per day with non-coffee drinkers.

All analyses were performed using R version 4.3.2.

## 3. Results

### 3.1. Sample Characteristics

The study sample consisted of 7420 individuals, with a mean age of 53.2 years (SD = 16.8), and 72.2% of them were women (Table 1). The majority of the participants were non-smokers (87.7%) and non-alcohol drinkers (88.6%). On average, participants reported consuming 0.21 cups of coffee per day (SD = 0.47). More than half of the sample did not drink coffee (54.0%) and only 2.7% reported consuming more than one cup per day. As shown in Appendix A, individuals with habitual coffee intake tended to be younger, male, ever-smokers, and ever-alcohol drinkers, and have higher education levels (all *p*-values < 0.001).

### 3.2. Phenome-Wide Association Analysis

A phenome-wide association analysis was performed to identify disease outcomes associated with coffee intake. Participants were followed for a median of 20.0 years for mortality (interquartile range = 17.2–21.4), and the median follow-up period for other disease outcomes ranged from 16.8 to 20.4 years. After adjusting for age, sex, BMI, smoking, alcohol drinking, and education, any coffee intake was significantly associated with a reduced risk of 10 outcomes at an FDR < 0.05 (Figure 1). These diseases were grouped into six categories, including mental disorders (dementias), sense organs (dizziness and giddiness), circulatory system (atrial fibrillation), respiratory (respiratory infections and painful respiration), symptoms (fever), and dermatologic (atopic dermatitis). The hazard ratios (HRs) for these outcomes ranged from 0.43 to 0.77, with the largest effect sizes observed for painful respiration (HR = 0.43; 95% confidence interval [CI] = 0.29–0.63), atopic dermatitis (HR = 0.53; 95% CI = 0.38–0.75), and delirium dementia (HR = 0.56; 95% CI = 0.40–0.78) (Table 2).

For the 10 outcomes associated with any coffee intake at FDR < 0.05, subgroup analysis was performed based on sex and age. Most of the associations were significant only in women and younger adults aged <60 years (Appendix A). However, we also observed a significant interaction between painful respiration and sex and age, where the association between any coffee intake and painful respiration was stronger among men and older adults aged ≥60 years (Appendix A).

### 3.3. Selected Disease and Mortality Outcomes

We further assessed the association between coffee intake and the specific disease and mortality outcomes identified from the literature [2]. Kaplan–Meier curves in Appendix A showed that, compared to non-coffee drinkers, individuals with a coffee intake of ≤1 cup and >1 cup per day had significantly higher survival probabilities (log-rank *p*-value < 0.001). In the adjusted Cox models, any coffee intake was significantly associated with a reduced risk of all-cause mortality (HR = 0.82; 95% CI = 0.73–0.93) and Alzheimer’s disease (HR = 0.49; 95% CI = 0.31–0.77) at FDR < 0.05 (Figure 2, Table 3, and Appendix A). The association between any coffee intake and all-cause mortality was largely consistent across age and sex subgroups (Table 3).

### 3.4. Sensitivity Analysis

The association between any coffee intake and the identified clinical outcomes remained essentially unchanged when excluding the first two years of follow-up (Appendix A). Moreover, when coffee intake was analyzed as a categorical variable in the phenome-wide analysis, the outcomes significantly associated with an intake of ≤1 cup of coffee per day were consistent with those associated with any coffee intake (Appendix A). However, none of the disease outcomes were associated with an intake of >1 cup of coffee per day at an FDR < 0.05 (Appendix A).

## 4. Discussion

Using a phenome-wide association approach, this is the first study comprehensively investigating the long-term effects of coffee intake on disease incidence and mortality in a Chinese population. Over a 20-year follow-up period, we found that any coffee intake was associated with a reduced incidence of painful respirations, dementia, atopic dermatitis, atrial fibrillation, respiratory infections, and dizziness (FDR < 0.05). Additionally, any coffee intake was significantly associated with a decreased risk of all-cause mortality. These results suggest that light coffee consumption may be beneficial to health.

In general, our study aligns with a growing body of evidence indicating positive impacts of coffee intake on health outcomes, rather than harmful effects [2]. In particular, coffee intake has been associated with a reduced risk of cardiovascular diseases [9,10], including atrial fibrillation [36,37], as well as a lower risk of dementia and cognitive decline [17,18]. Consistent with these findings, our study demonstrated a strong, inverse relationship between any coffee intake and the risks of atrial fibrillation and dementia. The potential cardiovascular and neuronal health benefits may be attributed to the presence of anti-oxidants in coffee, such as chlorogenic acid, trigonelline, and melanoidins, which may inhibit inflammation [38,39,40]. Likewise, recent Mendelian randomization studies have suggested that genetically predicted circulating caffeine levels are associated with a lower risk of obesity, type 2 diabetes, and osteoarthritis [41,42], which could be explained by lower chronic inflammation, improved lipid profiles, and altered protein and glycogen metabolism [42].

We also revealed a potential link between coffee intake and a lower incidence of respiratory diseases and symptoms, atopic dermatitis, and dizziness, all of which have been scarcely investigated in the literature. Several previous studies have reported a reduced risk of respiratory diseases, including asthma and chronic obstructive pulmonary disease, associated with coffee consumption [43,44]. Apart from its anti-inflammatory effects, caffeine in coffee is a mild bronchodilator, which may reduce respiratory muscle fatigue and potentially improve airway function [45]. Meanwhile, the lower risk of dizziness [46] may be explained by the blood pressure-lowering effects of the chlorogenic acids present in coffee [47,48], whereas the anti-inflammatory properties of coffee may be beneficial to atopic dermatitis [49], although these findings need to be confirmed by further studies.

In the secondary aim, we examined the association between coffee intake and the specific disease and mortality outcomes selected from the previous umbrella review [2]. Consistent with prior findings in European [20], Asian [22], and other non-white populations [50], we confirmed an inverse association between coffee intake and all-cause mortality in both men and women. Furthermore, we highlighted a significant association between any coffee intake and a lower risk of Alzheimer’s disease, which aligns with our phenome-wide results showing an inverse relationship between coffee intake and dementia.

Although we did not find evidence of a detrimental effect associated with coffee consumption, it is important to note that, due to the relatively low levels of coffee consumption in our sample (where over half of the participants were non-coffee drinkers), our analysis focused primarily on any coffee intake. We also did not observe a significant association between consuming >1 cup of coffee per day and any of the disease outcomes studied. While this could be due to limited statistical power in our analysis, several previous studies have also suggested potential J-shaped relationships between coffee intake and disease outcomes, where excessive consumption beyond a certain amount may not confer additional benefits and could even lead to an increased risk of certain conditions, such as stroke [51], heart failure [52], and hypertension [53]. Similarly, a previous study in the UK Biobank found that only light-to-moderate coffee consumption (0.5–3 cups per day), but not high coffee consumption (>3 cups/day), was significantly associated with a lower risk of mortality [21]. Therefore, future studies are warranted to explore the potential dose–response relationships between coffee intake and the identified clinical outcomes.

The existing literature was predominantly conducted in Caucasian populations, in which, on average, a high coffee intake was observed. Notably, in Caucasian populations, coffee consumption is known to be strongly correlated with smoking [26,27]. Thus, the observed association could be confounded—at least in part—by smoking, even after adjusting for smoking in the statistical models. In contrast, the prevalence of smoking in the Hong Kong Chinese population is considerably lower (only 12.3% were ever-smokers in our sample). Furthermore, it is known that genetic variations play an important role in caffeine metabolism and coffee consumption [54], and there is a large difference in genetic background between Caucasian and Chinese populations. Therefore, our study addressed the association between modest coffee intake and health outcomes in a unique population that is different from Caucasians in terms of lifestyle, amount of coffee consumption, and genetics.

Our study has important clinical implications. As coffee is a beverage that is widely consumed globally, it is of public health importance to evaluate the potential benefits and harms of coffee intake. Using high-quality data from a territory-wide EHR, we systematically evaluated the relationship of coffee intake with >400 medical conditions and uncovered several potential beneficial effects of coffee intake. Furthermore, we validated specific outcomes that were highlighted in a previous umbrella review [2], which, if replicated in diverse populations, could inform dietary guidelines for the general public and specific patient populations.

### Strengths and Limitations

The strengths of this study include the use of a validated coffee measurement [30] and access to electronic health record data spanning a relatively long follow-up period of 20 years, allowing us to investigate the long-term health impacts of coffee intake. In addition, most, if not all, of the medical conditions were ascertained by clinicians, which should have a high positive predictive value as shown in our previous validation studies [55,56].

However, we also acknowledge some limitations. Although we have controlled for several potential confounders and minimized the impact of reverse causation by excluding the first 2 years of follow-up in the sensitivity analysis, as in other observational studies, our results could still be biased by residual and unmeasured confounding. In addition, due to a lack of data on specific coffee types, we were unable to compare different coffee brewing methods and roasting times [57]. We also did not account for changes in coffee consumption over time due to a lack of data on coffee intake after baseline. Finally, as the data were collected over 20 years ago when coffee consumption in Hong Kong was less prevalent, our findings may not be generalizable to other populations with higher coffee consumption. Therefore, further studies in Asian populations are warranted to replicate our findings.

## 5. Conclusions

In this comprehensive analysis over a 20-year follow-up, we identified multiple health outcomes associated with coffee intake, including a lower incidence of several respiratory, mental, circulatory, and dermatologic diseases and symptoms. Moreover, in this population of Hong Kong Chinese adults, any coffee intake was associated with a 51% reduced risk of Alzheimer’s disease and an 18% lower risk of all-cause mortality. These findings provide new evidence on the long-term effects of coffee intake and suggest that light coffee consumption is safe and may also confer health benefits. Future studies should investigate the mechanisms underlying these associations and examine potential causal relationships.

## Figures and Tables

**Figure 1 nutrients-16-03536-f001:**
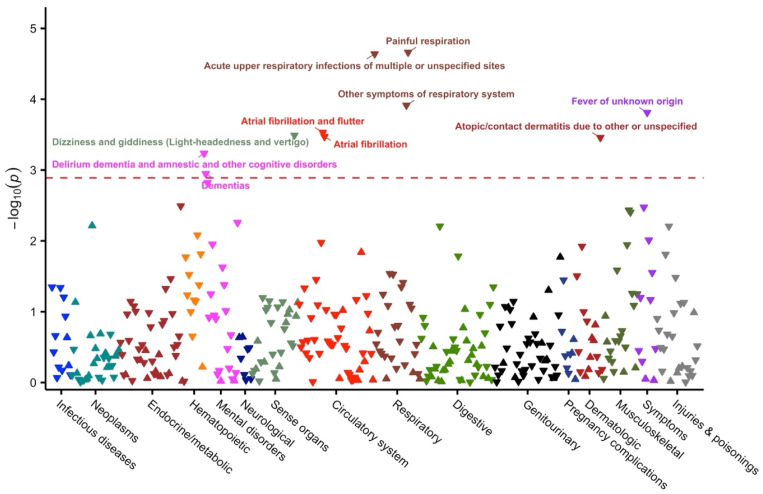
Manhattan plot for the phenome-wide analysis of any coffee intake on disease outcomes. Labeled symbols are the diseases significantly associated with any coffee intake at FDR < 0.05. The dotted red line indicates the significance threshold at FDR = 0.05. All estimates were calculated based on Cox models adjusted for age, sex, body mass index, smoking, alcohol drinking, and education. The up-pointing triangles indicate positive associations (hazard ratio > 1), whereas the down-pointing triangles indicate inverse associations (hazard ratio < 1).

**Figure 2 nutrients-16-03536-f002:**
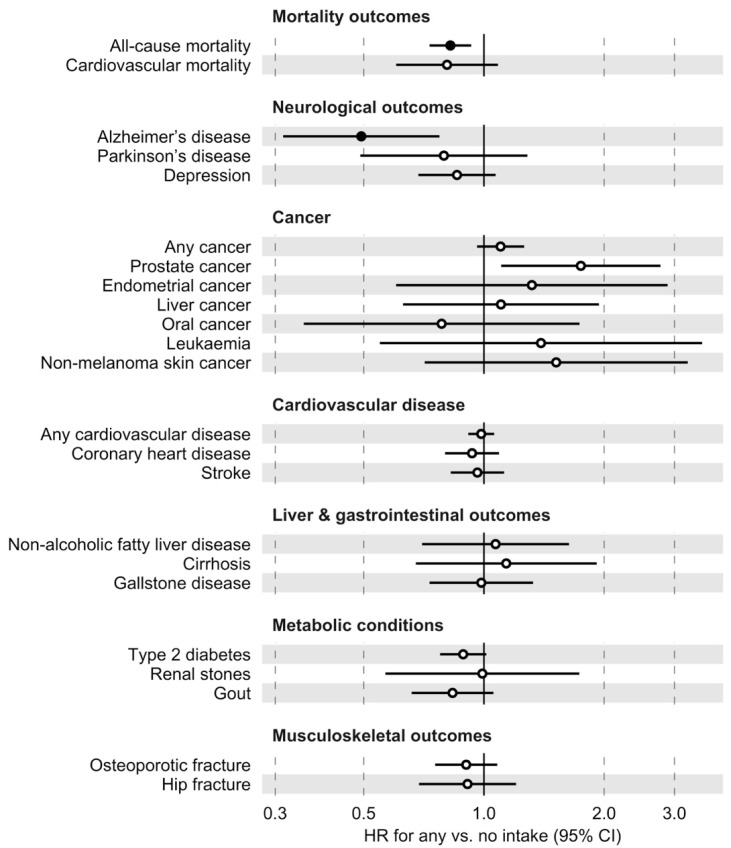
Association between any coffee intake and specific disease and mortality outcomes selected from the literature. All models were adjusted for age, sex, body mass index, smoking, alcohol drinking, and education. Filled circles indicate a false discovery rate < 0.05 and empty circles indicate non-significant estimates. CI, confidence interval; HR, hazard ratio.

**Table 1 nutrients-16-03536-t001:** Sample characteristics by sex.

Characteristic	Overall (n = 7420)	Women (n = 5355)	Men (n = 2065)
Age, mean (SD)	53.2 (16.8)	51.2 (16.3)	58.2 (16.9)
Body mass index, mean (SD)	22.8 (3.6)	22.6 (3.7)	23.3 (3.3)
Ever-smoker, n (%)	916 (12.3)	250 (4.7)	666 (32.3)
Ever-alcohol drinker, n (%)	849 (11.4)	302 (5.6)	547 (26.5)
Education level, n (%)			
Primary or below	2414 (32.5)	1899 (35.5)	515 (24.9)
Secondary	2983 (40.2)	2122 (39.6)	861 (41.7)
College or university	2023 (27.3)	1334 (24.9)	689 (33.4)
Coffee intake, cup/day, mean (SD)	0.21 (0.47)	0.18 (0.42)	0.29 (0.57)
Non-coffee drinker, n (%)	4006 (54.0)	2978 (55.6)	1028 (49.8)
≤1 cup of coffee per day, n (%)	3215 (43.3)	2283 (42.6)	932 (45.1)
>1 cup of coffee per day, n (%)	199 (2.7)	94 (1.8)	105 (5.1)
Died during follow-up, n (%)	1558 (21.0)	964 (18.0)	594 (28.8)

**Table 2 nutrients-16-03536-t002:** Fully adjusted association between any coffee intake and the top identified phecode diagnoses.

Phecode	Description	N ^a^	Incident Cases	HR (95% CI) ^b^	*p*-Value
Mental disorders					
290	Delirium dementia and amnestic and other cognitive disorders	6936	255	0.56 (0.40, 0.78)	5.8 × 10^−4^
290.1	Dementias	6933	250	0.57 (0.41, 0.80)	1.1 × 10^−3^
Sense organs					
386.9	Dizziness and giddiness (light-headedness and vertigo)	7212	943	0.77 (0.66, 0.89)	3.2 × 10^−4^
Circulatory system					
427.2	Atrial fibrillation and flutter	6816	481	0.67 (0.54, 0.83)	2.9 × 10^−4^
427.21	Atrial fibrillation	6803	467	0.67 (0.54, 0.83)	3.4 × 10^−4^
Respiratory					
465	Acute upper respiratory infections of multiple or unspecified sites	7284	588	0.68 (0.57, 0.81)	2.3 × 10^−5^
512	Other symptoms of respiratory system	7388	533	0.68 (0.56, 0.83)	1.2 × 10^−4^
512.2	Painful respiration	7027	152	0.43 (0.29, 0.63)	2.2 × 10^−5^
Symptoms					
783	Fever of unknown origin	7398	776	0.73 (0.62, 0.86)	1.5 × 10^−4^
Dermatologic					
939	Atopic/contact dermatitis due to other or unspecified origins	7261	167	0.53 (0.38, 0.75)	3.5 × 10^−4^

*Note:* The 10 listed phecodes are those associated with any coffee intake in the phenome-wide analysis at FDR < 0.05. ^a^ Number of individuals without the medical condition at baseline. ^b^ All models were adjusted for age, sex, body mass index, smoking, alcohol drinking, and education.

**Table 3 nutrients-16-03536-t003:** Association between any coffee intake and all-cause mortality.

Subgroup	N	Deaths	HR (95% CI) ^a^	*p*-Value
Full sample	7420	1558	0.82 (0.73, 0.93)	0.002
Women	5355	964	0.82 (0.70, 0.97)	0.019
Men	2065	594	0.81 (0.68, 0.98)	0.026
Age < 60 years	4498	184	0.95 (0.70, 1.29)	0.74
Age ≥ 60 years	2922	1374	0.80 (0.70, 0.91)	7.5 × 10^−4^

^a^ All models were adjusted for age, sex, body mass index, smoking, alcohol drinking, and education.

## Data Availability

The dataset used in the current study cannot be shared due to The Personal Data (Privacy) Ordinance (Cap. 486) in Hong Kong as stated in the consent form. For all requests regarding data, please contact the corresponding author at lung1212@hku.hk.

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
