# Peer review of "Phenome-Wide Analysis of Coffee Intake on Health over 20 Years of Follow-Up Among Adults in Hong Kong Osteoporosis Study"

_nutrients, 2024, doi:10.3390/nu16203536_

Round 1
Reviewer 1 Report
Comments and Suggestions for Authors
1. The introduction of this manuscript is relatively weak, and more recent research or systematic literature review should be supplemented.
2. For example, a 2020 review paper "Coffee, Caffeine, and Health" (DOI: 10.1056/NEJMra1816604) in the New England Journal of Medicine shows that caffeine intake may have health benefits, including improving the brain's cognitive abilities, reduce the risk of depression and Parkinson's disease, reduce liver cirrhosis, reduce the risk of liver cancer and endometrial cancer, reduce gallstones and urinary tract stones, reduce mortality, etc.
3. Did this study eliminate invalid survey samples, outliers and their proportions? If so, please add.
4. The quantitative data or numerical data of relevant indicators in this study are fully presented and explained.
5. The conclusion of this manuscript is a general narrative and lacks quantitative results. It is recommended to add "research limitations" and "further research plans", and may also appropriately supplement the results of reliability and validity analysis.
6. Overall, this study has value in academic, dietary intake and health aspects.
Author Response
Comment 1: The introduction of this manuscript is relatively weak, and more recent research or systematic literature review should be supplemented.
Response 1: Thank you very much for your time to review our manuscript, and the constructive feedback to help us improve our work. We have now largely revised the Introduction section, included more information about the potential health effects of coffee and its bioactive compounds such as chlorogenic acids and diterpenes, and cited more recent research and systematic reviews (pages 1-2, lines 35-57).
The added references include:
- van Dam, R.M.; Hu, F.B.; Willett, W.C. Coffee, Caffeine, and Health. N. Engl. J. Med. 2020, 383, 369–378, doi:10.1056/NEJMra1816604.
- Rojas-González, A.; Figueroa-Hernández, C.Y.; González-Rios, O.; Suárez-Quiroz, M.L.; González-Amaro, R.M.; Hernández-Estrada, Z.J.; Rayas-Duarte, P. Coffee Chlorogenic Acids Incorporation for Bioactivity Enhancement of Foods: A Review. Molecules 2022, 27, doi:10.3390/molecules27113400.
- Ren, Y.; Wang, C.; Xu, J.; Wang, S. Cafestol and Kahweol: A Review on Their Bioactivities and Pharmacological Properties. Int. J. Mol. Sci. 2019, 20, doi:10.3390/ijms20174238.
- Di Pietrantonio, D.; Pace Palitti, V.; Cichelli, A.; Tacconelli, S. Protective Effect of Caffeine and Chlorogenic Acids of Coffee in Liver Disease. Foods (Basel, Switzerland) 2024, 13, doi:10.3390/foods13142280.
- Torabynasab, K.; Shahinfar, H.; Payandeh, N.; Jazayeri, S. Association between Dietary Caffeine, Coffee, and Tea Consumption and Depressive Symptoms in Adults: A Systematic Review and Dose-Response Meta-Analysis of Observational Studies. Front. Nutr. 2023, 10, 1051444.
- Hong, C.T.; Chan, L.; Bai, C.-H. The Effect of Caffeine on the Risk and Progression of Parkinson’s Disease: A Meta-Analysis. Nutrients 2020, 12, doi:10.3390/nu12061860.
Comment 2: For example, a 2020 review paper "Coffee, Caffeine, and Health" (DOI: 10.1056/NEJMra1816604) in the New England Journal of Medicine shows that caffeine intake may have health benefits, including improving the brain's cognitive abilities, reduce the risk of depression and Parkinson's disease, reduce liver cirrhosis, reduce the risk of liver cancer and endometrial cancer, reduce gallstones and urinary tract stones, reduce mortality, etc.
Response 2: Following your suggestion, we have now added the 2020 review paper in the Introduction as reference #1. We have also added the potential health benefits of coffee intake accordingly: “Recent epidemiological studies have predominantly suggested a protective effect of coffee consumption against several diseases, including cardiovascular diseases [9,10], type 2 diabetes [11], stroke [12,13], liver diseases [14], depression [15], Parkinson’s disease [16], cognitive decline and dementia [17,18]. Additionally, coffee has been associated with reduced risks of specific cancers such as liver and endometrial cancer [19], as well as all-cause and cardiovascular mortality [20–22].” (page 2, lines 45-50)
Comment 3: Did this study eliminate invalid survey samples, outliers and their proportions? If so, please add.
Response 3: Thank you for pointing this out. The invalid study samples (i.e., those who have incomplete or missing data on the demographic and study variables) have been excluded from the study as detailed in the flowchart in Supplementary Figure S1 and in section 2.1 (page 2, lines 82-85): “For the present analysis, we excluded individuals who were not linked to the CDARS (n=188), with missing demographic data or aged <18 years (n=41), and with missing information on coffee consumption (n=1,306) or covariates (n=494), resulting in a sample size of n=7,420”.
We have indeed also checked for the distribution of coffee intake, which is highly skewed (as shown in Supplementary Figure S2). In particular, 43.3% of the sample had ≤1 cup of coffee intake per day, and only 199 individuals had >1 cup and 33 had >2 cups of coffee intake per day. If we use the usual definition of outliers (more than 1.5*IQR above the third quartile), then 1,312 individuals with coffee intake >0.4 cups per day will be excluded and led to a loss of information. Therefore, we decided to focus on any vs. no coffee intake and have not excluded outliers in the analysis. This has been discussed in the Methods section (page 3, lines 93-96): “Due to the overall low coffee consumption in our sample (Supplementary Figure S2), we considered coffee intake as a binary variable (any vs. no coffee intake) and examined the association between any coffee intake and the health outcomes in the analysis.”
Comment 4: The quantitative data or numerical data of relevant indicators in this study are fully presented and explained.
Response 4: Thank you very much for your comment and appreciation.
Comment 5: The conclusion of this manuscript is a general narrative and lacks quantitative results. It is recommended to add "research limitations" and "further research plans", and may also appropriately supplement the results of reliability and validity analysis.
Response 5: Thank you for your suggestion. We have now added the quantitative results and added some future directions in the conclusion section (page 9, lines 313-319): “Moreover, in this population of Hong Kong Chinese adults, any coffee intake was associated with a 51% reduced risk of Alzheimer’s disease and a 18% lower risk of all-cause mortality. These findings provide new evidence on the long-term effects of coffee intake and suggest that a light coffee consumption is safe and may also confer health benefits. Future studies should investigate the mechanisms underlying these associations and examine potential causal relationships.”
Research limitations have also been discussed under the section “Strengths and limitations” in the Discussion section (page 9, lines 292-308).
Comment 6: Overall, this study has value in academic, dietary intake and health aspects.
Response 6: Thank you very much for your overall positive feedback to our work!
Reviewer 2 Report
Comments and Suggestions for Authors
Thank you for the opportunity to review this manuscript. It presents an interesting study about the association between coffee intake and the incidence of diseases and mortality risk over 20 years among community-dwelling Chinese adults.
However, I have some concerns presented below.
General comments
P value in the whole manuscript must be expressed as p value.
Specific Comments
2. Materials and Methods
Line 57, please add the acronym: HKOS
At the end of the introduction you stated, line 48, that you aimed to investigate the long-term effects of coffee intake on health over 20 years in a cohort of community-dwelling Chinese adults. But in the methods section, you stated that coffee intake was assessed by a self-reported food frequency questionnaire administered during the baseline assessments, line 75. We suppose that baseline assessment was between 1995 and 2010, line 58. How is the way you have investigated the long-term effects over 20 years? I think it is the median data, because I have read this after in the results section, but I think that you have to clarify this issue a little more in the methods section, 2.1. Participants subsection, line 66.
Line 81. “we considered coffee intake as a binary variable (any vs. no coffee intake) in the main analysis, and further categorized it into non-coffee drinkers, ≤1 cup, and >1 cup of coffee intake per day in the sensitivity analysis.” Here you have to explain better to which analysis you refer with “main analysis” and “sensitivity analysis”. You have to think that the reader has not yet read the statistical analysis section and in the aims that is not explained: “We first performed a phenome-wide association analysis to identify disease outcomes associated with coffee intake. To validate previous findings, we further performed a targeted assessment of specific disease and mortality outcomes previously linked to coffee intake [2]”.
Statistical analysis section
Can you please relate the explanation you perform here with the aims of the study? Clearly, we do not know when we read this section the analysis performed for the secondary aim. The big problem is that we are not completely able to understand the methodology until we reach the results section. Thus, please, clarify here all the analysis you show in the results section.
3. Results
Line 145 and Figure 1. How is it possible that you speak in line 145 about reduced risk of 10 outcomes if in the figure 1, we see that almost all the Labeled symbols are pointing upwards? But it is true that in the table 1 we see all the HR are bellow 1. Please clarify the meaning of the symbols in figure 1. And review for all the Manhattan plots
Table 1. Legend. Please change 1 and 2 to a and b.
Line 176. “any coffee intake had a significantly higher” Please, correct and use the categorization you have used in the analysis and that you have presented in the figure.
Author Response
Comment 1: Thank you for the opportunity to review this manuscript. It presents an interesting study about the association between coffee intake and the incidence of diseases and mortality risk over 20 years among community-dwelling Chinese adults.
However, I have some concerns presented below.
Response 1: Thank you very much for time to review our manuscript and the constructive feedback to help us improve our work. Please find our responses to your concerns below.
Comment 2: P value in the whole manuscript must be expressed as p value.
Response 2: Thank you for your comment. We have now changed all P values as “p-value” throughout the manuscript (page 4, line 150; page 6, line 194; Table 2; Table 3).
Comment 3: Line 57, please add the acronym: HKOS
Response 3: Thank you for pointing this out. We have now added the acronym “HKOS” accordingly (page 2, line 67)
Comment 4: At the end of the introduction you stated, line 48, that you aimed to investigate the long-term effects of coffee intake on health over 20 years in a cohort of community-dwelling Chinese adults. But in the methods section, you stated that coffee intake was assessed by a self-reported food frequency questionnaire administered during the baseline assessments, line 75. We suppose that baseline assessment was between 1995 and 2010, line 58. How is the way you have investigated the long-term effects over 20 years? I think it is the median data, because I have read this after in the results section, but I think that you have to clarify this issue a little more in the methods section, 2.1. Participants subsection, line 66.
Response 4: Thank you for your comment, and we apologize for the confusion. While the assessment of coffee intake was done at baseline between 1995 and 2010, we followed the participants for a median of 20 years via electronic health records in CDARS to examine the impact of coffee intake on the incidence of health outcomes. We have now clarified this in the Methods, Participants subsection (page 2, lines 76-78): “Electronic health records of the HKOS participants have been retrieved from the CDARS until January 15, 2024, allowing us to follow the participants longitudinally with a median of 20 years.”
Comment 5: Line 81. “we considered coffee intake as a binary variable (any vs. no coffee intake) in the main analysis, and further categorized it into non-coffee drinkers, ≤1 cup, and >1 cup of coffee intake per day in the sensitivity analysis.” Here you have to explain better to which analysis you refer with “main analysis” and “sensitivity analysis”. You have to think that the reader has not yet read the statistical analysis section and in the aims that is not explained: “We first performed a phenome-wide association analysis to identify disease outcomes associated with coffee intake. To validate previous findings, we further performed a targeted assessment of specific disease and mortality outcomes previously linked to coffee intake [2]”.
Response 5: Thank you for your comment, and we apologize for the confusion. We referred the “main analysis” to the analysis that addressed both the first and second aims, and “sensitivity analysis” to an additional analysis to test the robustness of the findings (i.e., to test whether using a different definition of the exposure variable would affect the results) and to examine potential dose-response relationship.
For clarity, we have now removed the “sensitivity analysis” description from the “Habitual coffee intake” section: “Due to the overall low coffee consumption in our sample (Supplementary Figure S2), we considered coffee intake as a binary variable (any vs. no coffee intake) and examined the association between any coffee intake and the health outcomes in the analysis.” (page 3, lines 93-96).
Comment 6: Statistical analysis section. Can you please relate the explanation you perform here with the aims of the study? Clearly, we do not know when we read this section the analysis performed for the secondary aim. The big problem is that we are not completely able to understand the methodology until we reach the results section. Thus, please, clarify here all the analysis you show in the results section.
Response 6: Thank you for pointing this out. We have now revised the statistical analysis section to make it more clearly align with the two aims: “A phenome-wide association analysis was first performed using Cox models to identify disease outcomes (phecodes) associated with any coffee intake… To validate previous findings, we further assessed the association between any coffee intake and the incidence of the 19 specific diseases and two mortality outcomes selected from the literature [2]…Cox models were used to estimate the association between coffee intake and the specific disease and mortality outcomes, adjusted for age, sex, BMI, smoking, alcohol drinking, and education.” (page 3, lines 119-130)
Comment 7: Line 145 and Figure 1. How is it possible that you speak in line 145 about reduced risk of 10 outcomes if in the figure 1, we see that almost all the Labeled symbols are pointing upwards? But it is true that in the table 1 we see all the HR are bellow 1. Please clarify the meaning of the symbols in figure 1. And review for all the Manhattan plots
Response 7: Thank you for your comment. We have reviewed and double checked the Manhattan plots carefully, and the labelled symbols (triangles) in Figure 1 are indeed pointing to the correct directions (mostly pointing downwards, indicating a reduced risk). To make it clearer, we have updated the footnote of Figure 1 to specify that we are referring the down-pointing triangles to inverse associations: “The up-pointing triangles indicate positive associations (hazard ratio >1), whereas the down-pointing triangles indicate inverse associations (hazard ratio <1).”
Comment 8: Table 1. Legend. Please change 1 and 2 to a and b.
Response 8: Thank you for your comment. We have now changed all the legends of the tables to a and b.
Comment 9: Line 176. “any coffee intake had a significantly higher” Please, correct and use the categorization you have used in the analysis and that you have presented in the figure.
Response 9: Thank you for your comment. We have now corrected the sentence accordingly: “Kaplan-Meier curves in Supplementary Figure S4 showed that compared to non-coffee drinkers, individuals with a coffee intake ≤1 cup and >1 cup per day had significantly higher survival probabilities (log-rank p-value <0.001).” (page 6, lines 191-194)
Reviewer 3 Report
Comments and Suggestions for Authors
The authors aimed to investigate the long-term effects of coffee intake on health over 20 years in a cohort of community-dwelling Chinese adults
The introduction is short but clear to define the objective of the study.
The methodology is clearly described, although only the basal coffee intake is indicated. We do not know the subsequent evolution of coffee intake.
The results are expressed in an orderly manner and are very easy to understand.
The discussion is adapted to the results obtained by comparing them with the data obtained in the Caucasian population. The absence of control in the subsequent coffee intake is included in the limitations.
Author Response
Comment 1: The authors aimed to investigate the long-term effects of coffee intake on health over 20 years in a cohort of community-dwelling Chinese adults
The introduction is short but clear to define the objective of the study.
Response 1: Thank you very much for your time to review our manuscript and the constructive feedback to help us improve our work. As detailed in our response to the first comment of reviewer #1, we have now expanded the Introduction and added more information about the latest research on the health effects of coffee intake (pages 1-2, lines 35-57).
Comment 2: The methodology is clearly described, although only the basal coffee intake is indicated. We do not know the subsequent evolution of coffee intake.
Response 2: Thank you for your comment. This is indeed a limitation of our study, as we don’t have data on the subsequent evolution of coffee intake of the participants. We have now emphasized this point in the “Strengths and limitations” section on page 9, line 303-304: “We also did not account for changes in coffee consumption over time due to a lack of data on coffee intake after baseline.”
Comment 3: The results are expressed in an orderly manner and are very easy to understand.
Response 3: Thank you very much for your comment and appreciation.
Comment 4: The discussion is adapted to the results obtained by comparing them with the data obtained in the Caucasian population. The absence of control in the subsequent coffee intake is included in the limitations.
Response 4: Thank you for your comment and the overall positive feedback to our work!
Round 2
Reviewer 2 Report
Comments and Suggestions for Authors
I want to thank the authors for all the corrections performed. Special thanks for the clarification of the comment about the Labeled symbols of the Manhattan plots. I was confused because of the lines that emerge from the triangle. Now, I have understand that I have to look only the point of the triangle.